# A systematic review of disease related stigmatization in patients living with prostate cancer

Derek Larkin[1]☯*, Alison J. Birtle[2]‡, Laura Bradley[1]‡, Paola Dey[3]☯, Colin R. Martin[4]‡, Melissa Pilkington[5]‡, Carlos Romero-Rivas[6]‡

1 Department of Psychology, Edge Hill University, Lancashire, United Kingdom, 2 Department of Oncology, Rosemere Cancer Centre, Lancashire Teaching Hospitals, Preston, United Kingdom, 3 Faculty of Health, Social Care and Medicine, Edge Hill University, Lancashire, United Kingdom, 4 Institute for Clinical and Applied Health Research (ICAHR), University of Hull, Hull, United Kingdom, 5 Department of Psychology, Manchester Metropolitan University, Manchester, United Kingdom, 6 Department of Evolutive and Educational Psychology, AUM, Madrid, Spain

☯ These authors contributed equally to this work.
‡ AJB, LB, CRM, MP and CRR also contributed equally to this work.
* Derek.Larkin@edgehill.ac.uk

**Data Availability Statement:** All relevant data are within the manuscript and its Supporting Information files.

## Abstract

### Background

Prostate cancer has been shown to be susceptible to significant stigmatisation, because to a large extent it is concealable, it has potentially embarrassing sexual symptoms and has significant impact on the psychosocial functioning.

### Methods

This review included studies that focused on qualitative and/or quantitative data, where the study outcome was prostate cancer and included a measure of stigmatization. Electronic databases (CINAHL, Medline, PubMed, PsycInfo, Cochrane Library, PROSPERO, and the Joanna Briggs Institute) and one database for grey literature Opengrey.eu, were screened. We used thematic analysis, with narrative synthesis to analyse these data. We assessed risk of bias in the included studies using the RoBANS.

### Results

In total, 18 studies met review inclusion criteria, incorporating a total of 2295 participants. All studies recruited participants with prostate cancer, however four studies recruited participants with other cancers such as breast cancer and lung cancer. Of the 18 studies, 11 studies evaluated perceived or felt stigma; four studies evaluated internalised or self-stigma; three studies evaluated more than one stigma domain.

### Discussion

We found that patients living with prostate cancer encounter stigmatisation that relate to perception, internalisation, and discrimination experiences. We also identified several

**Funding:** This project received funding from Edge Hill University to undertake this research. The funders had no role in study design, data collection and analysis, decision to publish, or preparation of the manuscript.

**Competing interests:** The authors have declared that no competing interests exist.

significant gaps related to the understanding of prostate cancer stigmatization, which provides an opportunity for future research to address these important public health issues.

## Registration

This systematic review protocol is registered with PROSPERO, the international prospective register of systematic reviews in health and social care. Registration number: CRD42020177312.

## Introduction

Prostate cancer is the 4[th] most common cancer worldwide and the most common cancer in men, with over 130 cases diagnosed every day in the United Kingdom totalling 48,500 cases per annum [1]. The prostate is a small gland located below the bladder responsible for secreting one of the components of semen [2]. Several risk factors for prostate cancer have been identified, age is the most significant, along with family history, genetic factors, race, lifestyle and dietary habits [3, 4]. Although only about 1 in 350 men under the age of 50 years are diagnosed with prostate cancer, the rate increases to 1 in 52 for ages 50 to 59 years, 1 in 19 for ages 60 to 69 years, and 1 in 11 for men 70 years and older, which equals a lifetime risk of 1 in 8. Although some men are diagnosed after the cancer has spread beyond the prostate (40% of all English new prostate cancer cases), advances in treatment options ensure that 78% of men survive prostate cancer for 10 or more years ([5] 2013–2017 England & Wales data). As a consequence, however, men diagnosed with prostate cancer are living longer, some with debilitating treatment-related side-effects [6]. Given the possible health impacts of prostate cancer itself and the potential side effects of the treatment, one area that has received increasing research in prostate cancer is the role that stigma plays in psychosocial functioning.

The treatment pathways for individuals diagnosed with prostate cancer inevitably vary by individuals and by stage. Treatment intent can be either curable or palliative and prostate cancer can be heterogenous [7]. For example, active surveillance avoids unnecessary treatment of low-risk cancers; radical prostatectomy (removal of the prostate gland) and external radiotherapy are treatments usually given with curative intent [2]. Prostate cancer is driven by testosterone, where hormone therapy is given as the "backbone" of treatment in metastatic disease, or in combination with radiation in organ confined or locally advanced prostate cancer [7]. In the latter instance, it is given for 2–3 years after radiotherapy. In organ confined disease, it may be given for 6 months or so, and in metastatic disease, indefinitely [2]. The side effects of these treatments include lack of libido, weight gain, hot flushes, erectile dysfunction, and changes in mood [7]. Radiotherapy and prostatectomy can both cause long term changes in urinary and sexual function as well as bowel function. Some men may also experience cosmetic shortening of the penis and feel anxious about using a urinal rather than a stall [7]. However, even prior to a formal diagnosis, men may experience symptoms that in themselves are a potential source of stigma, for example needing to urinate more frequently often during the night, needing to rush to the toilet, difficulty in starting to urinate (hesitancy), straining or taking a long time while urinating, or having a weak flow [2].

While the aetiology of prostate cancer is not fully understood, there are several factors that may increase the risk of developing the condition. There is an increased risk among men of African-Caribbean or African descent [8], and those with a sibling or father who developed prostate cancer before the age of 60 or a close female relative who has developed breast cancer [7], there is increasing evidence which suggests poor lifestyle may be associated with prostate cancer. Obesity is associated with elevated incidence of prostate cancer [1, 9]. It is also

associated with a higher risk of aggressive prostate cancer, with higher tumour stage, and grade on biopsy [10]. There is also an association between a generally poor diet and lifestyle and the increased risk of prostate cancer [11, 12]. There is also limited evidence that a diet high in calcium may be linked to an increased risk of developing prostate cancer [7]. Although the evidence is limited, nutrients including fat, protein, carbohydrates, vitamins (vitamin A, D, and E), and polyphenols, potentially affect prostate cancer pathogenesis and progression through a mechanism including inflammation, antioxidant effects, and the action of sex hormones [13].

Given the possible health effects of prostate cancer and the potential side effects of the treatment, one area that has received increasing research interest in prostate cancer is in the role that stigma plays in psychosocial functioning. Chronic illness stigmatization has been explored across numerous health conditions, most notably in HIV/AIDS [14] mental health [15, 16], but has also been explored in cancers such as breast [17, 18], colorectal, skin [19] and lung cancer [20–22] as well as other chronic conditions such as obesity [23]. Illness stigma has been shown to have numerous health implications including limiting access to medical care, increasing treatment non-adherence, increasing psychological distress, decrease self-esteem and self-efficacy and increased illness symptoms [24].

In his seminal work, Goffman [25] defines stigma as a state of spoiled identity brought on by being deeply discredited and socially rejected for having a particular trait. According to Goffman, stigma, a Greek term, refers to bodily signs designed to expose something unusual and bad about the status of the signifier. Goffman [25] argued that stigmatised persons may be reduced in people's minds, from a whole and ordinary person to disgraced and discounted one. The stigmatised person may even be subject to discriminatory behaviour by others [20], and even reduced accessibility to diagnosis and treatment [26]. Since Goffman's initial conceptualisation of stigma, it has evolved; Link and Phelan [27], for example, state that stigma exists when interrelated components converge. The first of these components are the labels used to distinguish differences from one another. The second is the beliefs that the dominant culture hold to label others with undesirable characteristics. The third is to label individuals to accomplish some degree of separation of us and them. The fourth component is to use labels to establish status loss and discrimination. Finally, Link and Phelan [27] suggest that stigmatization is entirely contingent on access to social, economic and political power. This prevailing stigma theory can be delineated into three principal domains; 1) perceived or felt stigma, 2) internalised or self-stigma, and 3) enacted stigma, or actual discrimination [24, 27]. Using these domains, and knowing that, in many parts of the world, cancer continues to carry a significant amount of stigma, prostate cancer is therefore predisposed to illness related stigma.

Identifying the aetiological explanations for disease activity is essential but brings with it the potential for others to view prostate cancer as under the individuals' control, due to their apparent inability to manage their poor lifestyle. However, masculinity, sexual performance, and urinary dysfunction are characteristics of prostate cancer that may also attract the focus for stigmatization. The language used to describe prostate cancer treatment, coupled with the emasculating way in which the treatments are discussed in the media might influence how prostate cancer stigma is constructed [28]. To date, no review has explored disease related stigmatization in patients living with prostate cancer, utilising both quantitative and qualitative research. In the current review we evaluated the three primary stigma domains and their relationship to patient outcomes, and disease management.

## Method

The present study is reported following the Preferred Reporting Items for Systematic Reviews and Meta-analysis (PRISMA).

## Eligibility criteria

We included studies that comprised of men that had been diagnosed with prostate cancer as their primary condition, either curative or palliative. The phenomenon of interest for this review was 1) perceived or felt stigma, 2) internalised or self-stigma, and 3) enacted stigma, or actual discrimination [24, 27]. This review considered studies that focused on qualitative and/ or quantitative data, where the study outcome was on the topic of prostate cancer and included a measure of stigmatization. We excluded studies if they did not have prostate cancer and stigmatisation as a primary research question or outcome. The search was limited to studies conducted and published in English language between January 2000 and January 2021 to map onto the stigma model outlined by [24, 27].

## Information sources

Prior to the start of this review, a preliminary search including key terms (i.e., prostate, prostate cancer, stigma, and stigmatization) was performed to identify any similar reviews on the topic. It was established that no relevant reviews on the research question had been registered or published. The search was limited to studies conducted and published in English language between January 2000 and January 2021 to map onto the stigma model outlined by [24, 27].

A systematic search across 8 electronic databases (CINAHL, Medline, PubMed, PsycInfo, Cochrane Library, PROSPERO, and the Joanna Briggs Institute) and one database for grey literature (Opengrey.eu) was carried out from March 2020 to June 2020 and replicated December 2020, to January 2021. The search was developed and tested through an iterative process including two authors (DL and LB).

## Search strategy

The search strategy used a wide range of controlled vocabulary (MeSH terms) and keywords transferable across all databases. Vocabulary and syntax were adjusted for database requirements; and keywords were truncated to broaden results. Keywords included prostat*, PCa, PC, DRE, "digital rectal examination", "prostate specific antigen", PSA, oncolog*, cancer*, tumour*, tumor*, stigma*, blam*, prejudice, sham*, discrimin*, bull*, teas*.

## Selection process

Study titles and abstracts generated by the electronic search were screened for relevance (independently by authors DL & LB) and full text articles retrieved for a more detailed review. Reference lists of identified articles were reviewed for additional studies. Unpublished manuscripts, systematic reviews & meta-analysis, case studies and dissertations were not included in the review. Articles identified by the database searches were reviewed independently by the authors DL & LB (with a third review author acting as an arbiter if necessary) for relevance to prostate cancer and the stigma construct, those not addressing stigma (stigmatization) and prostate cancer were removed from the full review.

## Data collection process

Electronic databases yielded 5259 results in total including grey literature; reference list searches elicited 1 further study. 3335 studies were retained after duplicates were removed, then screened (title and abstract) for eligibility, of these 3313 were removed. Following full paper screening, by authors DL & LB, of the remaining 46 studies, a further 28 studies were excluded, for failing to fulfil the inclusion criteria 25 were excluded because they did not explicitly investigate stigma, and 1 was identified as a dissertation and finally 2 did not report

or analyse prostate cancer findings. No studies were excluded for having a high or unclear risk of bias [29]. In total, 18 studies were included in the final analysis.

## Data items

The phenomenon of interest for this review was 1) perceived or felt stigma, 2) internalised or self-stigma, and 3) enacted stigma, or actual discrimination [24, 27]. Of the 18 studies, 10 of which were qualitative studies and 8 quantitative (See Table 1). DL and LB carried out the data extraction independently on all selective studies using full study reports. Information was extracted on (1) Authors (2) Inclusion criteria (3) Number of participants (4) Research question (5) Study methods (i.e., questionnaires or interviews) (6) Measures (7) Patient/Partners Ages (8) Stigma Domain (9) Key findings. Any disagreements were resolved through discussion.

## Study risk of bias assessment

We assessed risk of bias in the included studies using the Risk of Bias Assessment tool for Non-randomized Studies (RoBANS) [30]. The RoBANS contains 6 domains including the selection of participants, confounding variables, measurement of intervention (exposure), blinding of outcome assessment, incomplete outcome data and selective outcome reporting. Two review authors (DL & LB) independently applied the tool to each included study and recorded supporting information and justifications for judgements of risk of bias for each domain (low; high; unclear). Any discrepancies in judgements of risk of bias or justifications for judgements were resolved by discussion to reach consensus between the two review authors, with a third review author acting as an arbiter if necessary. Following guidance given for RoBANS we derived an overall summary 'Risk of Bias' judgement (low; high; unclear) for each specific outcome, whereby the overall RoBANS for each study was determined by the highest RoBANS level in any of the domains that were assessed.

Critical appraisal of methodological quality of included papers were undertaken independently by two authors (DL & LB). The Joanna Briggs Institute Critical Appraisal Checklist for Qualitative Research tool was used for qualitative studies, and Effective Public Health Practice Project (EPHPP) quality assessment tool for quantitative studies.

## Synthesis methods

As a consequence of the wide variety of data, which included qualitative studies, and inconsistent reporting of quantitative data, a metanalysis was not possible, we therefore decided that a narrative synthesis constituted the best approach to synthesise the findings of the studies. Following Lisy and Porritt [31] structure a narrative synthesis was chosen to summarise the data as it allowed the data to be analysed and contextualized. Firstly, studies were identified according to the prevailing stigma domains. Of the 18 studies within this review 11 studies evaluated perceived or felt stigma; four studies evaluated internalised or self-stigma; three studies evaluated more than one stigma domain. Further synthesis was undertaken in the form of thematic analysis, which involved the searching of studies, listing and presenting results in a tabular form, subsequently narrative synthesis was conducted.

## Results

### Study selection

A total of 2295 participants were involved. All studies recruited participants with prostate cancer, but four studies also recruited participants with other cancers such as breast and lung

**Table 1. Study characteristics.** Data extracted from each of the reviewed journal articles.

| Author | Inclusion Criteria | No of Participants | Research Question | Study Method | Measures | Age Years | Stigma Domain | Key Findings |
|---|---|---|---|---|---|---|---|---|
| Allensworth-Davies et al. [35] | Men >50, USA resident Gay— localized PCa at least 1 year | 111 | Masculine self-esteem in gay men | Cross-sectional survey | SF-12, EPIC, MSES/ MASSS | 50–74 | Perceived or felt stigma | PCa care providers can reduce **stigma** by creating a supportive environment for gay man. |
| Arrington [39] | Not explicitly reported | 16 | What common themes arise in the illness experiences of prostate cancer survivors? | Interviews | Thematic Analysis | 66–81 | Internalised or self-stigma & | Survivors acknowledged the permanent, **stigmatizing** "cancer" label, but found PCa care providers as a source of information and social support groups as sources of information and emotional support. |
| Bamidele et al. [37] | Black African (BA) /Caribbean (BC) men, UK resident, >35, PCa at least 3months. Partners–No restrictions on ethnicity, age or gender. | 25 men & 11 partners | Access and recruitment barriers for BA & BC men and their partners in research | Interviews | Grounded Theory | Not Reported | Internalised or self-stigma | Increased engagement with healthcare professionals and gatekeepers could facilitate better access to Black African/Caribbean populations in research. Cultural awareness of the **stigma** of cancer in BA and BC communities, and the influence gatekeepers can have in controlling access to potential participants. |
| Broom [46] | Not explicitly reported: Men recruited via specific support group and local magazine advertisement | 33 | Cultural constructions of masculinity and experiences of PCa in Australian society | Interviews | Thematic Analysis | Not Reported | Perceived or felt stigma | Investigative, diagnostic, and treatment procedures pose significant difficulties, and **stigma** for many men, especially in relation to their idealised constructions of masculinity. |
| Else-Quest et al. [22] | Please see article. | 172 (46 with PCa) | Perceived stigma and self-blame associated with poor psychological adjustment | Predominantly questionnaires (one qualitative question) | SSGS, RSES, STAI (Anxiety), STAI (Anger), CES-D/ | 35 to 92 (PCa = M.72.89) | Perceived or felt stigma | Participants who reported internal causal attributions reported poorer psychological adjustment. Self-blame significantly mediated the link between perceived **stigma** and self-esteem and anxiety. |

*(Continued)*

**Table 1.** (*Continued*)

| Author | Inclusion Criteria | No of Participants | Research Question | Study Method | Measures | Age Years | Stigma Domain | Key Findings |
|---|---|---|---|---|---|---|---|---|
| Ernst et al. [32] | Men between 18 and 75 years, (ii) time of diagnoses not more than 30 months before and (iii) new diagnosis or relapse. | 858 (268) with PCa) | To what extent do cancer patients feel stigmatized & are there significant associations between the level of stigmatization and QoL? | Questionnaires | EORTC, QLQ-C30, SIS-D, PHQ-D | 18–75 | Perceived or felt stigma | Across all cancer sites, the dimensions of **stigmatization** were in the lower and middle range, with the highest values found for isolation. **Stigmatization** was lowest among prostate cancer patients. |
| Esser et al. [33] | Men between 18 and 75 years, (ii) time of diagnoses not more than 30 months before and (iii) new diagnosis or relapse. | 858 (268 with PCa) | To measure the effect of perceived stigmatization on depressive symptomatology | Questionnaires | SIS-D, PHQ-D, FKB-20 | 60.7 (mean) age | Perceived or felt stigma | Perceived **stigmatization** is an important and generalizable risk factor for depressive symptomatology among cancer patients. |
| Ettridge et al. [47] | A diagnosis or treatment for prostate cancer within the last 24 months, aged 18 years or older, proficient at English | 28 | Men's experiences of PCa, perceived stigma and self-blame, social isolation, unmet need and help-seeking. | Interviews | Thematic Analysis | 28–82 | Perceived or felt stigma & Enacted stigma, or actual discrimination | Perceived **stigma** was associated with prostate cancer and cancer in general, which sometimes acted as a barrier to disclosure. Descriptions of emotional distress, social isolation and anxiety demonstrate the impact of prostate cancer. |
| Gray et al. [44] | Men with PCa. Married or living with partner, English speakers. Prostatectomy as their treatment choice but had not yet had surgery. | 34 couples | Decisions to share information (or not) with others about their diagnosis and ongoing medical situation. | Interviews | Numerical Unstructured Data Indexing Searching and Theorizing software | Men 50–68, Partners 42–72 | Perceived or felt stigma & Internalised or self-stigma | Factors related to limiting disclosure included men's low perceived need for support, fear of **stigmatization**, the need to minimize the threat of illness to aid coping, practical necessities in the workplace, and the desire to avoid burdening others. |
| LoConte et al. [34] | Patients with stage IV lung, breast, or prostate cancer, fluent in English | 172 (46 with PCa) | Levels of guilt and shame among patients with non–small-cell lung cancer (NSCLC) compared with breast and prostate cancer | Questionnaires | STAI, CES-D SSGS PCRS | 35–87 (56–87 with PCa) | Perceived or felt stigma | Patients with non–small-cell lung cancer had higher levels of perceived cancer-related **stigma** than patients with prostate cancer or breast cancer but not higher baseline levels of shame and guilt. Smoking is correlated with higher levels of guilt and shame. |

(*Continued*)

**Table 1.** (Continued)

| Author | Inclusion Criteria | No of Participants | Research Question | Study Method | Measures | Age Years | Stigma Domain | Key Findings |
|---|---|---|---|---|---|---|---|---|
| Maharaj and Kazanjian [45] | Men who are diagnosed with prostate cancer and fluent in English | 20 | Explore issues of intimacy and sexuality from the perspective of men with prostate cancer | Interviews | Thematic Analysis | 54–81 | Perceived or felt stigma & Internalised or self-stigma | Patients describe their psychosocial needs and experiences of personal loss and interpersonal loss, vulnerability, **stigma,** and self-blame |
| McConkey and Holborn [36] | Gay men with PCa | 8 | Explore the lived experience of gay men with prostate cancer | Interviews | Giorgi's phenomenological method | 47–66 | Perceived or felt stigma | Gay men with prostate cancer have unmet information and supportive needs–In relation to sexual dysfunction associated rehabilitation–issues associated with heteronormativity, minority stress, and **stigma** |
| Nelson et al. [43] | Newly diagnosed PCa, in an intimate and committed relationship. | 18 couples | Explore social support for men and their partners receive and provide in the first 12 months following PCa diagnosis | Interviews–over 3 times periods. | Thematic Analysis | 50–79 | Internalised or self-stigma | **Stigma** was identified to have a role in men's disclosure decisions. Partners generally provide high levels of support. Social support groups were highlighted as an important source of support. |
| Rising et al. [40] | PCa patients diagnosed with <5 years, With localised PCa. | 149 | The relationship between perceived stress, perceived cancer related stigma, weak-ties support preference and online community use for social support. | Questionnaires | GMPS / HIV Stigma Scale. W/STS | 40–85 | Perceived or felt stigma | Positive relationship between **stigma** and perceived stress in those who used online community for advice and emotional support. |
| Wagland et al. [38] | Men 18 to 42 months post diagnosis identified through cancer charities in England Wales Northern Ireland and hospital activity data in Scotland and invited by the treatment centre to complete a postal questionnaire–respondents were invited to interview. | 14 | Explore adjustment strategies adopted by Black African and Black Caribbean men in response to the impact of PCa diagnosis and treatment | Interviews | Framework analysis. | 55–85 | Internalised or self-stigma | Patient-centred care requires cultural sensitivity and interventions that challenge **stigma** and men's reluctance to disclose problems associated with prostate cancer and its treatment. |

*(Continued)*

**Table 1.** (Continued)

| Author | Inclusion Criteria | No of Participants | Research Question | Study Method | Measures | Age Years | Stigma Domain | Key Findings |
|---|---|---|---|---|---|---|---|---|
| Wood et al. [41] | PCa Survivors that were currently in romantic or intimate relations >18 years of age. | 85 | Explore the influence of stigma on prostate cancer survivor's quality of life. | Questionnaires | SIS, FACT-P | 56–75 | Perceived or felt stigma | PCa **stigma** has a significant negative influence on quality of life–No statistically significant difference for **stigma** based on demographic variables for example base and age. |
| Wood et al. [42] | PCa patients that were currently in romantic or intimate relations >18 years of age. | 80 couples | Explored the influence of stigma on PCa survivor's quality of life stigma and relationship satisfaction | Questionnaires | SIS, FACT-P, FACT-GP, CSI | 56–75 | Perceived or felt stigma | **Stigma** had a negative association with quality of life but not in relationship satisfaction |
| Yang et al. [48] | PCa patients T3 or T4 stage | 175 | Patient stigma, self-efficacy and anxiety mediated the relationship between doctor's empathy and cellular immunity. | Questionnaires + Peripheral venous blood samples. | SIS, CBI-B, HADS. T & NK cell count | Mean age 61.28 | Perceived or felt stigma | Clinical staff should focus on improving their empathy toward patients. Interventions that focus on patients' anxiety, **stigma,** and self-efficacy may be helpful to improve immunity. |

cancer [22, 32–34]. Two studies exclusively explored the lived experiences of gay men [35, 36] and two studies exclusively explored the lived experiences of black men [37, 38]. Of the 18 studies reviewed, seven were conducted in the United States [22, 34, 35, 39–42], three in the United Kingdom [37, 38, 43], two in Canada [44, 45], two in Germany [32, 33], two in Australia [46, 47], one in China [48] and one in the Republic of Ireland [36]. The age range of participants was 28 [47] to 92 [22]. See Tables 1 & 2 for study characteristics, and Fig 1 for the Preferred Reporting Items for Systematic review and Meta-Analysis (PRISMA) flow diagram of the literature searches.

## Risk of bias in studies

We used the RoBANS [30] to access risk of bias of the included studies. A summary of these assessments is provided in Table 3. In terms of overall risk of bias there were concerns about risk for the majority of studies, with three of these assessed at high risk of bias [37, 38, 44]. A text summary is provided in Table 3 of the six individual components of the risk of bias.

Domain and theme structure is shown in Fig 2.

## Domain 1: Perceived or felt stigma

Experiences of stigmatisation were noted to various degrees in all the reviewed studies. Perceived or felt stigma, however, was the domain that received the most attention within the body of the literature. Perceived stigma is the fear of being discriminated against, or the fear of an active stigma which arises from societies beliefs [27, 49].

**Table 2. Quantitative measures.**  Data extracted from the reviewed journal articles.

| Abbreviation | Questionnaire title | Reference |
| --- | --- | --- |
| SF-12 | The 12-Item Short Form Health Survey | Ware Jr, Kosinski and Keller [67] |
| EPIC | Expanded Prostate cancer Index Composite | Wei et al. [68] |
| MSES | Masculine Self-Esteem Scale | Clark et al. [69], Clark et al. [70] |
| SSGS | The State Shame and Guilt Scale | Marschall, Sanftner and Tangney [71] |
| RSES | Rosenberg's Self-esteem inventory | Rosenberg [72] |
| STAI (Anxiety) | Spielberger State-Trait Anxiety Inventory | Spielberger and Gorsuch [73] |
| STAI (Anger) | Spielberger State-Trait Anger Inventory | Spielberger et al. [74] |
| CES-D | Centre for Epidemiological Studies Depression Scale | Radloff [75] |
| EORTC QLQ-C30 | European Organization for Research and Treatment of Cancer | Hinz, Singer and Brähler [76] |
| SIS-D | Social Impact Scale (German) | Eichhorn, Mehnert and Stephan [77] |
| PHQ-D | Patient Health Questionnaire (German) | Gräfe et al. [78] |
| FKB-20 | German Body Image Questionnaire (Fragebogen zum Körperbild, | Albani et al. [79] |
|  | Cancer Severity | Deimling et al. [80] |
|  | Current Illness Symptoms (Cancer Related): | Armer et al. [81] |
|  | Functional Limitations | Nagi [82] |
|  | Recent Life Events | Kahana, Fairchild and Kahana [83] |
| GMPS | Global Measure of Perceived Stress | Cohen et al. [61] |
| HIV stigma Scale |  | Berger et al. [55] |
| W/STS | Weak-tie/Strong-tie Support network preference scale | Wright and Miller [84] |
| SIS | Social Impact Scale | Fife and Wright [50] |
| FACT-P | Functional Assessment of Cancer Therapy (Prostate) | Esper et al. [85] |
| FACT-GP | Functional Assessment of Cancer Therapy (General Population) | Cella et al. [86] |
| CSI | Couples Satisfaction Index | Funk and Rogge [87] |
| CBI-B | Cancer Behaviour Infantry (Brief version) | Heitzmann et al. [88], Merluzzi et al. [89] |
| HADS | Hospital Anxiety and Depression Scale | Zigmond and Snaith [90] |
| MASSS | MacDonald and Anderson social stigma scale | MacDonald and Anderson [51] |
| PCRS | Perceived cancer related stigma | LoConte et al. [34] |

**Quality of life.**  This review identified five studies that explored quality of life in relation to perceived or felt stigma in prostate cancer [32, 33, 41, 42, 48]. Each used the social impact scale (SIS), a self-report questionnaire that measures negative social attitudes. The SIS was developed by Fife and Wright [50]: the scale has four types of stigma corresponding to social rejection, financial insecurity, internalised shame and social isolation. Fife and Wright [50] suggest that the four subscales of the SIS can be separated into two main types of stigma: experiences of rejection and stigma, and social psychological feelings regarding stigma. Using the SIS to evaluate the influence of stigma on quality of life, Wood et al. [41] found that stigma negatively impacted well-being and quality of life for prostate cancer patients regarding financial insecurity and social isolation. Wood et al. [42] investigated the influence of stigma on quality of life concerning relationship satisfaction for prostate cancer survivors and their partners and found that the stigma faced by the patients had a significant negative association with quality of life, which also negatively affected their partner, but had little effect on the strength of their relationship. Ernst et al. [32] explored stigma and quality of life across four major cancer sites, prostate, breast, colon, and lung cancer. They found relatively low levels of stigmatisation, across all dimensions of the SIS, but stigma was most extensive among breast cancer patients. In prostate cancer patients, stigma was found to be a significant predictor of quality of life, however, across individual dimensions, lung cancer patients reported the highest levels of internalised shame which were significantly higher than the scores for prostate cancer patients.

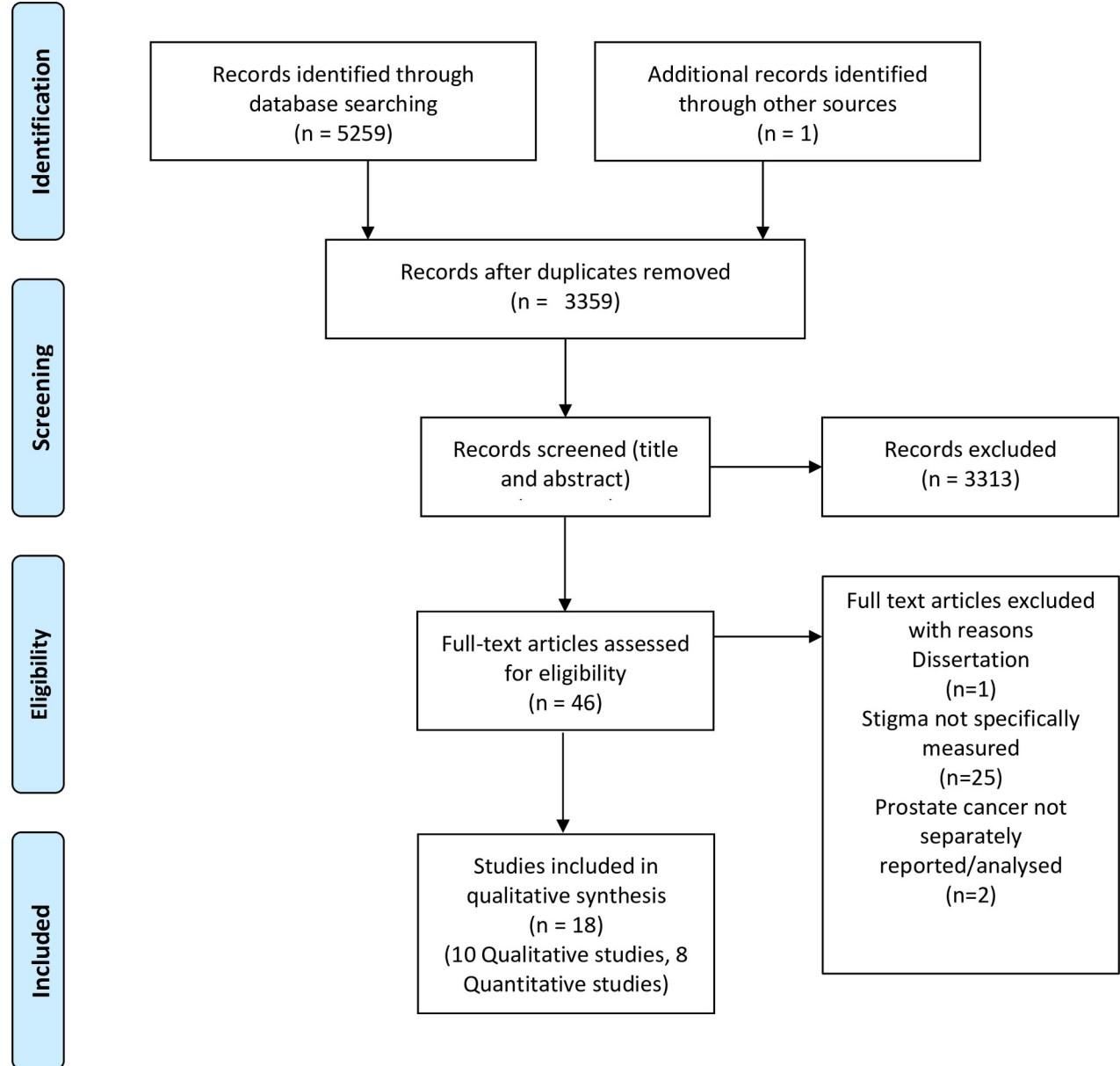

**Fig 1. PRISMA flow diagram for systematic review of stigmatization in prostate cancer.**

Esser et al. [33] investigated the effect of stigmatisation on depression in cancer patients using the SIS (same participants and protocol as Ernst et al. [32]. They report that across all four cancer sites stigmatisation showed total effects on depressive symptomology across all stigma dimensions. Yang et al. [48] however found a mediating factor that may influence perceived stigma in prostate cancer patients and improve quality of life. They report that patients exhibited significant increased anxiety and stigma and significantly reduced self-efficacy at 14 days after hospitalisation. However, at three months the psychological indicators had significantly improved. They found that the doctors' empathy directly affected patient self-efficacy, stigma, and anxiety. Low levels of empathy shown by the doctors resulted in poor scores for stigma and anxiety but a positive relationship for self-efficacy.

Table 3. The risk-of-bias assessment tool for nonrandomized studies (RoBANS).

| | Selection of participants | Confounding variables | Intervention (exposure) measurement | Blinding of outcome assessment | Incomplete outcome data | Selective outcome reporting | SUMMARY ASSESSMENT |
|---|---|---|---|---|---|---|---|
| | Selection bias caused by inadequate selection of participants | Selection bias caused by inadequate confirmation and consideration of confounding variable | Performance bias caused by inadequate measurement of intervention (exposure) | Detection bias caused by inadequate blinding of outcome assessment | Attrition bias caused by inadequate handling of incomplete outcome data | Reporting bias caused by selective outcome reporting | Risk of Bias |
| Allensworth-Davies et al. [35] | High | Low | Unclear | Unclear | Low | High | Low |
| Arrington [39] | Low | Unclear | Low | Low | Low | High | Low |
| Bamidele et al. [37] | Low | Low | High | Low | Low | Low | High |
| Broom [46] | Low | High | Low | Low | Low | High | Low |
| Else-Quest et al. [22] | High | Low | Low | Low | Low | High | Low |
| Ernst et al. [91] | Low | High | Low | Low | Low | High | Low |
| Esser et al. [33] | Low | High | Low | Low | Low | High | Low |
| Ettridge et al. [47] | Low | High | Low | Low | Low | High | Low |
| Gray et al. [44] | High | Unclear | Unclear | Low | High | High | High |
| LoConte et al. [34] | Low | Unclear | Low | Low | Low | High | Low |
| Maharaj and Kazanjian [45] | High | Unclear | Unclear | Low | Low | High | Low |
| McConkey and Holborn [36] | High | Low | Low | Low | Low | Low | Low |
| Nelson et al. [43] | High | Low | Low | Low | Low | High | Low |
| Rising et al. [40] | High | Low | Low | Low | Low | High | Low |
| Wagland et al. [38] | High | High | Unclear | Low | High | High | High |
| Wood et al. [41] | High | Low | Low | Unclear | Low | Unclear | Low |
| Wood et al. [42] | Unclear | Low | Low | Unclear | Low | Low | Low |
| Yang et al. [48] | High | Low | Low | Low | Low | High | Low |

**Masculinity.** This review identified three studies that explored masculinity in relation to perceived stigma in prostate cancer patients [35, 36, 46]. Using the McDonald and Anderson social stigma scale [51], Allensworth-Davies et al. [35] show that a significant minority (18%) of their participants reported severe levels of stigmatisation in relation to their prostate cancer, and that older men (75 and older) were significantly more likely to report severe stigmatisation than younger men. They also report that there was a strong association between replacing sex with other activities, severe stigma in the last month and masculine self-esteem. They report that on average, gay men who reported replacing sex with other activities also reported masculine self-esteem scores nearly 20 points lower than men who did not.

A systematic review of disease related stigmatisation in patients
living with prostate cancer

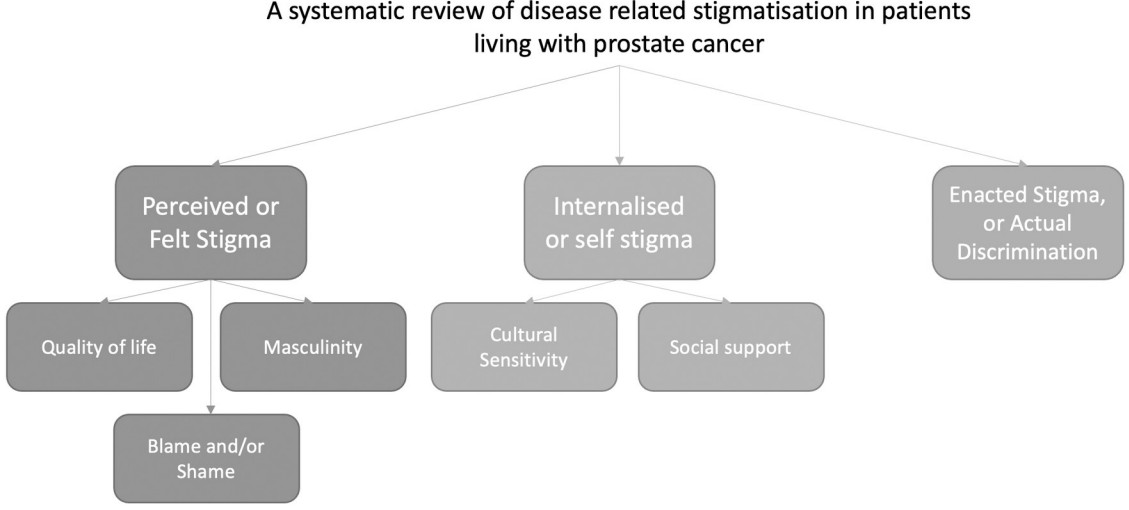

**Fig 2. Theme structure, demonstrating the stigma domains and developed themes.**

Using a qualitative descriptive design involving semi-structured interviews, McConkey and Holborn [36] asked gay men with prostate cancer to describe their treatment and diagnosis, their psychosocial needs, and experiences. Patient narratives demonstrate that although prostate cancer was a considerable source of stigmatisation, by far the greatest contributor was related to insults to their masculinity, sexual dysfunction, and associated rehabilitation. These insults negatively impacted on their quality of life, together with issues associated with heteronormativity, minority stress as a consequence of being gay with prostate cancer. One man describing prostate cancer as an assault on his masculinity, which was further complicated by the general public's perception of gay men being less masculine than heterosexual men. McConkey and Holborn [36] argue that this therefore significantly impacted on their willingness to interact with health services.

Broom [46] interviewed heterosexual men who also reported that prostate cancer was a substantial source of stigmatisation and an insult on their masculinity. Men reported difficulties with the diagnostic nature of some procedures (digital rectal examination) and in treatment decision-making the potential reduction in their ability to perform idealised forms of masculinity.

**Blame and/or shame.** This review identified three studies that explored blame and/or shame within the domain of perceived stigma in prostate cancer patients [22, 34, 45]. Else-Quest et al. [22] developed a single questionnaire item to assess cancer patients' (lung, breast, and prostate cancer) sense of stigmatization or blame for his or her cancer and asked participants to rate their agreement with the statement 'People judge me for my type of cancer'. They found that stigma negatively correlated with self-esteem but positively correlated with self-blame, anxiety, anger, and depressed affect, across all cancer types. Lung cancer was attributed to specific behaviours, whereas stigma associated with prostate cancer as a self-inflicted illness was related to a cancer diagnosis and not obviously related to a particular lifestyle or behaviour [22].

LoConte et al. [34] developed a six-item measure to assess self-blame related to cancer encompassing feelings of guilt, shame, and embarrassment. Items were subsequently averaged to create a scale of perceived cancer related stigma. Patients with breast or prostate cancer were significantly less likely to feel ashamed of their cancer compared to those with non-small

cell lung cancer (NSCLC). Prostate cancer patients were less likely to agree with the statement 'I am ashamed of my type of cancer' then patients with NSCLC, and less likely to agree with the statement 'my behaviour contributed to my cancer'. Prostate cancer patients were also less likely to report a history of being treated for depression or anxiety. Overall prostate cancer patients scored lower on the perceived cancer related stigma scale than did NSCLC patients.

In using a qualitative descriptive design involving semi-structured interviews Maharaj and Kazanjian [45] explored concerns with regard to stigmatization and shame-inducing aspects of the prostate cancer experience. They found that men tended to blame themselves or remain silent about their experiences, particularly in relation to erectile dysfunction and sexual health. They report that men tended to talk indirectly by describing experiences of other men whom they knew rather than sharing their own personal experiences.

## Domain 2: Internalised or self-stigma

This review identified five studies that explored the domain of internalised or self-stigma in relation to prostate cancer [37–40, 43]. Internalised or self-stigma is reported to be related to the poorest health outcome of the three stigma domains. In this domain individuals apply negative attitudes and stereotypes to themselves rather than rejecting them as false [24]. Individuals who internalise societal attitudes suffer from internalised or self-stigma, which has a variety of negative effects Corrigan, Watson and Barr [52]. Self-stigmatisation erodes one's sense of self-worth, undermining one's ability to achieve goals. As a result, the harm caused by self-stigma manifests itself first through an intrapersonal mechanism, rather than through poor health outcomes [53]. Stigmatised persons may internalise perceived prejudices and develop negative feelings about themselves [54].

**Social support.**   This review identified three studies that specifically explored social support in relation to internalised or self-stigma [39, 40, 43]. Using thematic analysis, Arrington [39] reports that prostate cancer patients feel less agency over their lives than non-stigmatised individuals. Patients acknowledged diminished ability to drive, to play golf, to leave the house for a long time, and to perform sexually. The authors, however, acknowledged that their findings did not mirror that of previous research regarding patients withdrawing from interactions with others, or being treated differently because of their disease. Patients instead identified as being uncertain upon diagnosis, subsequently identifying as information seekers. The narrative describes the physician as information source, an important form of social and emotional support.

Rising et al. [40] used the global measure of perceived stress and 11 items of the HIV stigma scale [55] to measure personalised stigma in order to evaluate prostate cancer related stigma, perceived stress, and social support. They report that there was a positive relationship between stigma and perceived stress. This relationship, however, was moderated by online community use for social support. Analyses also revealed a positive relationship between stigma and perceived stress in those who sought advice or emotional support from online groups. The authors suggest that men who feel stigmatized and are hesitant to seek help from family or friends may need extra guidance from their health care providers.

Using semi-structured interviews and thematic analysis, Nelson et al. [43] investigated how men with prostate cancer and their partners utilise social support. It was reported that although stigma was not cited as a direct cause of social isolation, it was a significant barrier in men's disclosure decisions, and contributed to their partners' inability to reveal prostate-related issues to others which almost certainly contributed to the level of social isolation they reported experiencing. Nelson et al. [43] show that in the male-female dyad, the female partners generally found it difficult to discuss concerns with close friends, therefore the female

partners reported feeling distressed, anxious and alone. Nelson et al. [43] reported that while social support groups were a significant source of support for men, their female partners did not receive the same emotional support and were required to manage not only their own anxiety but also their partners' discomfort. Although stigma was not cited as a direct cause of social isolation, the male partners' inability to reveal prostate-related issues to others appears to contribute to the high level of social isolation they both reported experiencing.

**Cultural sensitivity.** This review identified two studies that specifically explored internalised or self-stigma and cultural sensitivities in black African (BA) and black Caribbean (BC) men diagnosed with prostate cancer [37, 38] Wagland et al. [38] adopted semi-structured interviews and framework analysis and report that many of their participants stated that they kept their diagnosis to a small circle of family and friends, often not telling close relatives. Wagland et al. [38] argue that the reluctance to reveal may have caused some participants to avoid seeking clinical help for problems such as erectile dysfunction or to participate with prostate cancer support groups when directed to them by health professionals. Additionally, Wagland et al. [38] contend that black African and black Caribbean men seek minimal assistance from health professionals and receive the majority of support from mothers, friends, and churches. They report that nondisclosure was often associated with fear of stigma and the correlation of prostate cancer with erectile dysfunction and ideas of masculinity among participants. Wagland et al. [38] state that while men may initially be hesitant to reveal their diagnosis to others due to concerns about stigma and damaged masculinity, findings suggest that some men reframe their sense of manhood, shifting their focus away from sexuality and toward preserving their self and social identity by warning others about the condition.

Bamidele et al. [37] used grounded theory to explore the experiences of black African and black Caribbean men and their partners and the psychosocial needs after prostate cancer treatment. This study was principally an exploration of the barriers and facilitators to recruiting black African and black Caribbean men to prostate cancer research. They report that recruitment barriers comprised of gatekeepers (cancer support groups, specifically the lead contact) and the stigma associated with prostate cancer disclosure. They report that nondisclosure was attributed to perceptions of self and social stigma associated with being diagnosed with prostate cancer, within the black African and black Caribbean cultural settings. Bamidele et al. [37] argue that cultural perceptions of prostate cancer such as fatality and emasculation often impact on black men's attitudes and behaviours towards public exposure of disease. They advocate that increased engagement with healthcare professionals and gatekeepers could facilitate better access to black African and black Caribbean populations.

## Domain 3: Enacted stigma, or actual discrimination

Overall, enacted stigma or actual discrimination is rarely reported in the literature, this may be because few studies have actively explored this stigma domain. Enacted stigma, or actual discrimination, refers to episodes of discrimination against individuals with a societally or culturally stigmatized condition solely on the ground of an apparent imperfection [56]. However, this reviewed identified one study which reports incidences of enacted stigma or actual discrimination [47].

Ettridge et al. [47] explored enacted stigma and report that some men have been accused of not looking after themselves, and were guilty of not acting sooner, when the symptoms first arose. The authors report that there was some suggestion of a stigma associated with prostate cancer as a self-inflicted illness which appeared to be more related to having a diagnosis of cancer rather than a diagnosis of prostate cancer. Ettridge et al. [47] report while trying to explain the effects of their prostate cancer, patients would be met with a reluctance to engage.

## Discussion

This review identified 18 studies that investigated the effects of stigmatization in relation to prostate cancer, the findings of which suggest that health-related stigma is part of the prostate cancer experience. Findings suggest that patients with prostate cancer are vulnerable to disease-related stigma, which manifests primarily as threats to their quality of life, masculinity, blame and/or shame, social support, and cultural sensitivities. As a result, it seems reasonable to conclude that stigma plays a significant role in the lives of men with prostate cancer and those who care for them.

Within the domain of perceived or felt stigma, challenges to quality of life, manifested as internalised shame, social rejection, social isolation, and financial insecurities, were reported [32, 33, 41, 42, 48]. The findings indicate that stigma has a detrimental effect on the quality of life of those with prostate cancer; however, when compared to those with breast cancer, the level of stigma was comparatively low, with lung cancer patients exhibiting the highest level of internalised shame. Regardless of the severity of stigma, it was discovered to be a major indicator of quality of life [32]. However, it was disclosed that empathy shown by clinical staff has a significant mediating effect on perceived stigma, which appears to result in improved quality of life, and lower levels of anxiety [48].

Masculinity was identified as a potential source of stigma; in gay men, this manifested as a reluctance to disclose their sexual orientation in predominantly heteronormative clinical settings, out of embarrassment associated with having a diagnosis involving the genital area and the associated sexual and urinary side effects [35, 36]. For heterosexual males, the attack on their masculinity came in the form of diagnostic procedures, especially digital rectal examinations, and treatment decisions, as well as the possibility of a decline in their ability to perform idealised forms of masculinity, with particular regard to sexual performance [46].

In terms of internalised shame, it was documented that prostate cancer patients may blamed themselves for their experiences, especially regarding erectile dysfunction and sexual health [38]. However, prostate cancer patients were less likely to report a history of being treated for depression or anxiety, compared to lung and breast cancer patients [34]. Prostate cancer patients also scored lower on the perceived cancer related stigma scale than did lung patients and were less likely to agree with the statement 'I am ashamed of my type of cancer' than patients with lung cancer, and less likely to agree with the statement 'my behaviour contributed to my cancer' [34]. Thus, it appears that, while prostate cancer patients bear a lesser burden of shame and blame compared to other cancer patients, it is still a significant burden.

Internalized or self-stigma in relation to prostate cancer is a significant factor, according to studies within this review. Internalized or self-stigma incorporate social support and cultural sensitivities [39, 40, 43]. Prostate cancer patients who are stigmatised, frequently experience a loss of agency and are hesitant to seek support from family and friends, preferring instead to seek support and guidance from healthcare providers and support groups [39]. However, the support networks did not extend to female partners, who did not receive the same level of emotional support and were required to manage not only their own anxiety, but also their partners' discomfort [43].

Studies also established that black African and Caribbean men living in the UK, preferred to keep their diagnosis to a small circle of family and friends and were unlikely to seek professional assistance for issues such as erectile dysfunction or join prostate cancer support groups [38]. Nondisclosure was associated with fear of stigma, associated with erectile dysfunction and participants' concepts of masculinity [37]. Given the critical nature of prostate cancer issues for black men, it is surprising that these subjects have received so little attention. To date, the limited social science research on prostate cancer has focused on large

epidemiological studies aimed at identifying factors associated with poor group outcomes and on persuading black men to participate in various medical and screening programmes. Both perspectives contribute to a unified view of black men. The danger inherent in these approaches is stereotyping, the establishment of social representations that confine individual black men to a framework that is unlikely to adequately represent their experiences.

Enacted or actual discrimination is the least well understood domain, yet it appears to have several potential detrimental effects on the patient, for example some men were deliberately evaded, or were accused of not looking after themselves [47]. Others tended to avoid the topic because it relates to social norms regarding sex. Enacted stigma or actual discrimination has been noted in previous research exploring the topic of cancer [see for example, 57–59], however there were limited examples within the studies of this review As a consequence of the scarcity of studies on enactive stigma in prostate cancer patients, additional research needs to be conducted in order to better understand this aspect of stigma, which appears to be a source of discrimination and to explore how active stigma influences health outcomes.

In 2020 a framework in which six major descriptions for men's current prostate cancer survivorship experience emerged: dealing with side effects; challenging; medically focused; uncoordinated; unmet needs; and anxious [60]. There was a total of 26 survivorship elements determined across six domains: health promotion and advocacy; shared management; vigilance; personal agency; care coordination; and evidence-based survivorship interventions. Stigmatisation is not directly addressed in the framework, however, topics raised within the current review are, such as psychosocial functioning, masculinity, social support, and quality of life. Stigmatisation therefore maps across a number of the domains within Dunn et al. [60] framework, clearly demonstrating the pervasive and encompassing nature of prostate cancer stigmatisation.

## Limitations

There was a possibility of publication bias in those only studies that specifically explored stigmatization as an outcome measure were included in this review, meaning that studies that report finding stigmatization as a secondary outcome, but did not have stigma as a principal research question were not included.

It is important to quantify stigma and its related causes to make an effective judgement of stigmatisation regardless of population. Given the enormous impact of stigma on numerous facets of life for men with prostate cancer, researchers require scales that accurately capture their specific lived experiences. However, stigma-related concepts can only be quantified using a few available scales that assess various aspects of stigma. Studies within this review utilised several stigma scales, for example, Global Measure of Perceived Stress [Perceived stress scale, 61], and the HIV Stigma Scale [55: Adapted for prostate cancer]; McDonalds and Anderson social stigma scale [designed for Rectal Cancer Paitents, 51], and the social stigma scale [SIS, 50]. The SIS was used by five studies within the review [32, 33, 41, 42, 48] to measure stigma in prostate cancer patients but was designed and validated for use in patients with depression, schizophrenia, or HIV/AIDS [62]. As a result, none of the studies included in this review used prostate cancer-specific stigma questionnaires, and none of the qualitative studies employed the same experimental design, even though all the studies examined prostate cancer stigma. There is no reason however, to believe this invalidates the results in each study, or the results of this review, but simply highlights the difficulties when measuring the highly complex topic of stigmatisation within any specific population.

An additional limitation was a reliance on Western literature, many of the studies reported in this review were conducted in Western countries, therefore the voice of Eastern or other

cultural settings is missing. This apparent bias in the literature could be accounted for by the incidence rate reported in different cultures for prostate cancer. A man living in North America is far more likely to be diagnosed with prostate cancer then a man living in South Central Asia for example [63]. Hsing, Tsao and Devesa [64] hypothesised that differences in incidence and mortality rates reported for numerous countries could be the result of underdiagnosis, underreporting, disparities in screening practices, disparities in health-care access, gaps in knowledge and awareness, and attitudes toward prostate cancer and associated screening, and cancer stigma. This last point is a very important issue, since for example in African countries such as Nigeria, patients with cancer who faced stigma were more likely to conceal their diagnosis and seek medical care later, while cancer stigma primarily resulted in adverse psychosocial outcomes for patients Akin-Odanye and Husman [65]. In addition, experiences related to stigmatization may change across cultures. For example, in a study conducted in Karnataka, India, breast or cervical cancer stigma was defined in terms of both actual (enacted) stigma, such as seclusion or verbal stigma, and perceived (fear of) stigma, in the event that a cancer diagnosis was reported [66]. But, in short, there is a gap in the scientific literature about how cancer-related stigmatization is modulated across cultures, and on how interventions could be carried out in non-Western cultures.

## Conclusion

This systematic review is the first to comprehensively review disease related stigmatization in patients living with prostate cancer, encompassing both qualitative and quantitative studies. Based on the results of this review, prostate cancer has been shown to be susceptible to significant stigmatisation, because to a large extent it is concealable, it has potentially embarrassing sexual symptoms and has potentially significant impact on the psychosocial functioning.

Stigma has been a growing concern in cancer literature, and this study aimed to illuminate how prostate cancer stigmatisation relates to the lives of individuals experiencing the condition. Based on this research, there is clear rationale for further research exploring factors that influence or impede quality of life for prostate cancer patients, as a consequence of stigma.

## Supporting information

**S1 Checklist. PRISMA 2020 checklist.**
(DOCX)

## Author Contributions

**Conceptualization:** Derek Larkin.

**Data curation:** Derek Larkin.

**Formal analysis:** Derek Larkin, Laura Bradley, Paola Dey.

**Funding acquisition:** Derek Larkin, Paola Dey, Colin R. Martin, Melissa Pilkington, Carlos Romero-Rivas.

**Investigation:** Derek Larkin, Paola Dey.

**Methodology:** Derek Larkin, Paola Dey, Carlos Romero-Rivas.

**Project administration:** Laura Bradley.

**Writing – original draft:** Derek Larkin, Laura Bradley, Paola Dey, Colin R. Martin, Melissa Pilkington, Carlos Romero-Rivas.

**Writing – review & editing:** Alison J. Birtle, Paola Dey, Colin R. Martin, Melissa Pilkington, Carlos Romero-Rivas.

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
