## [Decision Letter · Decision Letter 0]

29 Oct 2021

PONE-D-21-26750A Systematic Review of Disease Related Stigmatization in Patients Living with Prostate CancerPLOS ONE

Dear Dr. Larkin,

Thank you for submitting your manuscript to PLOS ONE. After careful consideration, we feel that it has merit but does not fully meet PLOS ONE’s publication criteria as it currently stands. Therefore, we invite you to submit a revised version of the manuscript that addresses the points raised during the review process.

We look forward to receiving your revised manuscript.

Kind regards,

Henry Woo

Academic Editor

PLOS ONE

Journal Requirements:

2. Thank you for stating the following in the Acknowledgments / Support Section of your manuscript: 

This research was supported by Research Investment Fund (RIF) Edge Hill University. 

DL received funding from Edge Hill University to undertake this research. The funders had no role in study design, data collection and analysis, decision to publish, or preparation of the manuscript.

Reviewers' comments:

Reviewer's Responses to Questions

**Comments to the Author**

1. Is the manuscript technically sound, and do the data support the conclusions?

Reviewer #1: Yes

Reviewer #2: Yes

2. Has the statistical analysis been performed appropriately and rigorously? 

Reviewer #1: Yes

Reviewer #2: N/A

3. Have the authors made all data underlying the findings in their manuscript fully available?

Reviewer #1: Yes

Reviewer #2: Yes

4. Is the manuscript presented in an intelligible fashion and written in standard English?

Reviewer #1: Yes

Reviewer #2: Yes

5. Review Comments to the Author

Reviewer #1: Well constructed and written paper on an important topic.

On page 3 the statement combining watchful waiting and active surveillance is not quite accurate. Watchful waiting is applicable to all cancer risk groups.

Reviewer #2: Please update referencing style as per journal guidelines.

Abstract:

Please do not begin your background paragraph with a statement about your methodology.

Introduction:

“Although some men are diagnosed after the cancer has spread beyond the prostate, (40% of all English new prostate cancer cases), advances in treatment options ensure that 78% of men survive prostate cancer for 10 or more years (2013-2017 England & Wales data).”

What are the references for these stats?

The second half of the introduction is informative and well-written.

Methods:

I enjoyed reading the methods section and the journey to final analysis and narrative-style review. Well done for a clear and transparent account of your search strategy and methods.

Results and Discussion:

Very little to fault. I have thoroughly enjoyed reading this review and am better informed as a result.

I would suggest presenting the 3 major themes and sub-themes in a pictorial or easily accessible format as a figure so that readers can get a grasp of your findings rapidly. This will be the most effective and efficient way to get your message across.

The second suggestion is to consider where your findings surrounding stigmatisation in prostate cancer fits into the survivorship essentials framework https://bjui-journals.onlinelibrary.wiley.com/doi/full/10.1111/bju.15159

Limitations:

I am interested in the authors’ thoughts on stigmatisation around this chronic disease in a predominantly western culture and literature compared with an eastern or other cultural setting.

Overall, I commend the authors on tackling this oft overlooked aspects of a very common disease.

6. PLOS authors have the option to publish the peer review history of their article (what does this mean?). If published, this will include your full peer review and any attached files.

Reviewer #1: No

Reviewer #2: **Yes: **Isaac A Thangasamy

---

## [Author Response · Author response to Decision Letter 0]

29 Nov 2021

Response to Reviewers

 We would like to thank the editorial team for taking the time and effort in considering our manuscript. We would particularly like to thank the two reviewers for their insightful comments. We have now taken on board all the comments from both reviewers and addressed each issue in turn below, without exception we have accepted and adopted the recommended changes. We now feel that the manuscript has been fundamentally improved, we therefore sincerely hope that with the changes made to the manuscript it can be once again considered for publication in PLOS ONE

1. Response: Manuscript now meets PLOS ONE style requirements. Both the main body of the text and author affiliations have been updated. 

Larkin, D*1 ¶Birtle, A. J2&, Bradley, L1&, Dey, P3¶, Martin, C.R4&, Pilkington, M5&, and Romero-Rivas, C6&

1 Department of Psychology, Edge Hill University, Lancashire, UK

2 Department of Oncology, Rosemere Cancer Centre, Lancashire Teaching 

 Hospitals, Preston, UK

3 Faculty of Health, Social Care and Medicine, Edge Hill University, Lancashire, UK

4 Institute for Clinical and Applied Health Research (ICAHR), University of Hull, Hull, UK

5 Department of Psychology, Manchester Metropolitan University, Manchester, UK

6 Department of Evolutive and Educational Psychology, AUM, Madrid, Spain 

*Corresponding author information: 

E-mail: Derek.Larkin@edgehill.ac.uk

¶These authors contributed equally to this work

& These authors also contributed equally to this work

2. Thank you for stating the following in the Acknowledgments / Support Section of your manuscript: 

This research was supported by Research Investment Fund (RIF) Edge Hill University. 

2A: Please remove any funding-related text from the manuscript and let us know how you would like to update your Funding Statement. Currently, your Funding Statement reads as follows: 

2A. Response: This funding statement has been removed from the manuscript. Please change the funding statement to the following text. 

“This project received funding from Edge Hill University to undertake this research. The funders had no role in study design, data collection and analysis, decision to publish, or preparation of the manuscript.”

2B. Please include your amended statements within your cover letter; we will change the online submission form on your behalf. 

2B. Response: The following text now appears in the cover letter. “This project received funding from Edge Hill University to undertake this research. The funders had no role in study design, data collection and analysis, decision to publish, or preparation of the manuscript.”

3. Please note that in order to use the direct billing option the corresponding author must be affiliated with the chosen institute. Please either amend your manuscript to change the affiliation or corresponding author or email us at plosone@plos.org with a request to remove this option.

3. Response: It’s our understanding that Edge Hill University has an agreement with PLOS ONE so that Edge Hill University will partly or fully pay the fees as a member of PLOS institutional accounting program The corresponding author (Dr Derek Larkin) is a full-time employee of Edge Hill University. 

4. Response: The captions have been added to both the tables and the figure. Labels have been labelled as Table 1, 2 & 3 Figure 1 & 2 These have I’ve also been updated within the text.

5. Response: To the best of the authors knowledge none of the papers have been retracted and none of the references have been removed, we have however included a subsequent reference in response to reviewers’ comments. References have been updated to conform to PLOS ONE citations, using the downloadable plug-in for Endnote. 

Additional references added in response to reviewer’s comment. 

60. Dunn J, Green A, Ralph N, Newton RU, Kneebone A, Frydenberg M, et al. Prostate cancer survivorship essentials framework: guidelines for practitioners. BJU International 2020. doi:10.1111/bju.15159.

63. Taitt HE. Global trends and prostate cancer: a review of incidence, detection, and mortality as influenced by race, ethnicity, and geographic location. Am J Men's Health. 2018;12(6):1807-23. doi:10.1177/1557988318798279.

64. Hsing AW, Tsao L, Devesa SS. International trends and patterns of prostate cancer incidence and mortality. Int J Cancer. 2000;85(1):60-7. doi:10.1002/(SICI)1097-0215(20000101)85:1<60::AID-IJC11>3.0.CO;2-B.

65. Akin-Odanye EO, Husman AJ. Impact of stigma and stigma-focused interventions on screening and treatment outcomes in cancer patients. Ecancermedicalscience. 2021. doi:10.3332/ecancer.2021.1308.

66. Nyblade L, Stockton M, Travasso S, Krishnan S. A qualitative exploration of cervical and breast cancer stigma in Karnataka, India. BMC women's health. 2017;17(1):1-15. doi:10.1186/s12905-017-0407-x.

Comments to the Author

6 Is the manuscript technically sound, and do the data support the conclusions?

Reviewer #1: Yes

Reviewer #2: Yes

Response: no response required

7. Has the statistical analysis been performed appropriately and rigorously? 

Reviewer #1: Yes

Reviewer #2: N/A

Response: no response required

8. Have the authors made all data underlying the findings in their manuscript fully available?

Reviewer #1: Yes

Reviewer #2: Yes

Response: no response required

9. Is the manuscript presented in an intelligible fashion and written in standard English?

Reviewer #1: Yes

Reviewer #2: Yes

Response: no response required

 10. Review Comments to the Author

Reviewer #1: Well-constructed and written paper on an important topic.

10A On page 3 the statement combining watchful waiting and active surveillance is not quite accurate. Watchful waiting is applicable to all cancer risk groups.

10A Response: Thank you for this helpful comment we have now removed the phrase “active surveillance” from the sentence. 

10B Reviewer #2: Please update referencing style as per journal guidelines.

10B Response: References have been updated to conform to PLOS ONE citations, using the downloadable plug-in for Endnote. 

10C Abstract:

Please do not begin your background paragraph with a statement about your methodology.

10C Response: The sentence “We systematically review the scientific literature on stigma as it relates to prostate cancer across three principal domains: perception, internalisation, and discrimination experiences.” Has now been removed from the background paragraph. 

10D Introduction:

“Although some men are diagnosed after the cancer has spread beyond the prostate, (40% of all English new prostate cancer cases), advances in treatment options ensure that 78% of men survive prostate cancer for 10 or more years (2013-2017 England & Wales data).”

What are the references for these stats?

10D Response: These stats came from cancer research UK https://www.cancerresearchuk.org/health-professional/cancer-statistics/statistics-by-cancer-type/prostate-cancer

Additional reference added in response to reviewer’s comment. 

5. Cancer Research UK. Prostate Cancer Statistics 2021 [cited 2021 August]. Available from: https://www.cancerresearchuk.org/health-professional/cancer-statistics/statistics-by-cancer- type/prostate-cancer

The second half of the introduction is informative and well-written.

Methods:

I enjoyed reading the methods section and the journey to final analysis and narrative-style review. Well done for a clear and transparent account of your search strategy and methods.

Results and Discussion:

Very little to fault. I have thoroughly enjoyed reading this review and am better informed as a result.

Response: no response required

10E I would suggest presenting the 3 major themes and sub-themes in a pictorial or easily accessible format as a figure so that readers can get a grasp of your findings rapidly. This will be the most effective and efficient way to get your message across.

10E Response: Thank you for this helpful suggestion we have now designed a figure that outlines the three major themes Labelled Figure 2

10F The second suggestion is to consider where your findings surrounding stigmatisation in prostate cancer fits into the survivorship essentials framework https://bjui-journals.onlinelibrary.wiley.com/doi/full/10.1111/bju.15159

10F Response: Many thanks for this suggestion we have considered the framework and have added an extra passage linking the framework to the current review. 

“In 2020 a framework in which six major descriptions for men's current prostate cancer survivorship experience emerged: dealing with side effects; challenging; medically focused; uncoordinated; unmet needs; and anxious [60]. There was a total of 26 survivorship elements determined across six domains: health promotion and advocacy; shared management; vigilance; personal agency; care coordination; and evidence-based survivorship interventions. Stigmatisation is not directly addressed in the framework, however, topics raised within the current review are, such as psychosocial functioning, masculinity, social support, and quality of life. Stigmatisation therefore maps across a number of the domains within Dunn et al. [60] framework, clearly demonstrating the pervasive and encompassing nature of prostate cancer stigmatisation.” 

10G Limitations:

I am interested in the authors’ thoughts on stigmatisation around this chronic disease in a predominantly western culture and literature compared with an eastern or other cultural setting.

10G Response: Would like to thank the reviewer for the suggestion. We have added a small section within the limitations section of the paper, discussing the lack of literature on prostate cancer stigmatisation across different cultures. 

“An additional limitation was a reliance on Western literature, many of the studies reported in this review were conducted in Western countries, therefore the voice of Eastern or other cultural settings is missing. This apparent bias in the literature could be accounted for by the incidence rate reported in different cultures for prostate cancer. A man living in North America is far more likely to be diagnosed with prostate cancer then a man living in South Central Asia for example [63]. Hsing, Tsao and Devesa [64] hypothesised that differences in incidence and mortality rates reported for numerous countries could be the result of underdiagnosis, underreporting, disparities in screening practices, disparities in health-care access, gaps in knowledge and awareness, and attitudes toward prostate cancer and associated screening, and cancer stigma. This last point is a very important issue, since for example in African countries such as Nigeria, patients with cancer who faced stigma were more likely to conceal their diagnosis and seek medical care later, while cancer stigma primarily resulted in adverse psychosocial outcomes for patients Akin-Odanye and Husman [65]. In addition, experiences related to stigmatization may change across cultures. For example, in a study conducted in Karnataka, India, breast or cervical cancer stigma was defined in terms of both actual (enacted) stigma, such as seclusion or verbal stigma, and perceived (fear of) stigma, in the event that a cancer diagnosis was reported [66]. But, in short, there is a gap in the scientific literature about how cancer-related stigmatization is modulated across cultures, and on how interventions could be carried out in non-Western cultures.”

10H Overall, I commend the authors on tackling this oft overlooked aspects of a very common disease.

Response: no response required

---

## [Editor Report · Decision Letter 1]

6 Dec 2021

A Systematic Review of Disease Related Stigmatization in Patients Living with Prostate Cancer.

PONE-D-21-26750R1

Dear Dr. Larkin,

We’re pleased to inform you that your manuscript has been judged scientifically suitable for publication and will be formally accepted for publication once it meets all outstanding technical requirements.

Kind regards,

Henry Woo

Academic Editor

PLOS ONE

---

## [Editor Report · Acceptance letter]

9 Dec 2021

PONE-D-21-26750R1 

A Systematic Review of Disease Related Stigmatization in Patients Living with Prostate Cancer. 

Dear Dr. Larkin:

I'm pleased to inform you that your manuscript has been deemed suitable for publication in PLOS ONE. Congratulations! Your manuscript is now with our production department. 

Kind regards, 

on behalf of

Prof. Henry Woo 

Academic Editor

PLOS ONE